# Sediment budget analysis of the Guayas River using a process-based model

Pedro D. Barrera Crespo[1,3,4], Erik Mosselman[1,2], Alessio Giardino[2], Anke Becker[2], Willem Ottevanger[2], Mohamed Nabi[2], and Mijail Arias-Hidalgo[3]

[1]Delft University of Technology, Delft, the Netherlands
[2]Deltares, Delft, the Netherlands
[3]Escuela Superior Politécnica del Litoral, ESPOL, Facultad de Ingeniería en Ciencias de la Tierra, Campus Gustavo Galindo Km 30.5 Vía Perimetral, P.O. Box 09-01-5863, Guayaquil, Ecuador
[4]HIDRODICON, Cuenca, Ecuador

*Correspondence to:* Pedro D. Barrera Crespo (pdbc.1985@gmail.com)

**Abstract.** The Equatorial Daule and Babahoyo rivers meet and combine into the tidal Guayas River, which flows into the largest estuary on the Pacific coast of South America. The city of Guayaquil, located along the Guayas, is the main port of Ecuador but, at the same time, the planet's fourth most vulnerable city to future flooding due to climate change. Sedimentation, which has increased in the recent years, is seen as one of the factors contributing to the risk of flooding. The cause of this sedimentation is subject of the current research. We used the process-based Delft3D FM model to assess the dominant processes in the river and the effects that past interventions along the river and its estuary have had on the overall sediment budget. Additionally, a simulation including sea level rise was used in order to understand the possible future impact of climate change on the sediment budget. Results indicate an increase in tidal asymmetry due to land reclamation and a decrease of episodic flushing by river floods due to upstream dam construction. These processes have induced an increased import of marine sediment potentially responsible for the observed sedimentation. This is in contrast with the local perception of the problem, which ascribes sedimentation to deforestation in the upper catchment. Only the deposition of silt and clay in connected stagnant water bodies could perhaps be ascribed to upstream deforestation.

## 1 Introduction

The morphodynamic evolution of rivers and estuaries is the result of the interaction between hydrodynamic conditions, channel geometry and sediment availability. Considerable progress has been made towards a better understanding of the processes governing the morphodynamic development of tidal rivers and estuaries and the possible impact of external anthropogenic interventions. Analytical models (e.g., Savenije, 2006; Van Rijn, 2011; Winterwerp, 2013) and numerical models (e.g., Van der Wegen and Roelvink, 2012; Giardino et al., 2014, 2018; Dam et al., 2016) have been developed in order to better explain and quantify these physical processes (Hoitink et al., 2017). Nevertheless, many estuaries and rivers around the world are still characterized by very limited knowledge on the overall morphodynamic behaviour, partly due to the lack of data for implementation and validation of complex numerical models.

The Guayas River in Ecuador is a typical example of a river which, despite its economic importance, is characterized by morphodynamic developments not yet clearly understood. Currently, a general sedimentation trend in the river has been observed, which has accelerated over the last decades. This sedimentation is locally perceived as a consequence of interventions carried out in the upper basin and natural events such as El Niño. Studies substantiating this perception are lacking. Moreover, the response to anthropogenic interventions carried out within the estuary and their potential effects on tidal dynamics have not yet been considered. The objective of this paper is to gain a better understanding on the overall sediment transport patterns and sediment budget in the river and its estuary and the interventions which may have affected it. This is essential in order to be able to propose possible adaptation measures to counteract the ongoing sedimentation processes.

## 1.1 Study area

Guayaquil, "La Perla del Pacífico" (The Pearl of the Pacific), is the most populated city (2,644,891 inhabitants) (INEC, 2017) and the industrial and commercial capital of Ecuador. The city owes its economic relevance to its location on the Guayas River, which in the past was considered the main seaborne trading route for the country.

The river is formed at the confluence of the Babahoyo and Daule rivers and is part of a larger estuarine system at the Gulf of Guayaquil, which is considered the largest estuary on the Pacific coast of South America (13,711 $km^2$) (Armijos and Montolío, 2008). Figure 1 shows the location and the extent of the study area.

The outer estuary extends from the mouth of the Gulf up to the Puna Island, about 130 km inland. The inner estuary includes the Estero Salado estuary and the Guayas River. The latter extends from the northern shore of the Puna Island up to the area subject to tidal influence, 50 km inland through the Babahoyo and Daule rivers.

The El Palmar islet, located at the confluence of the Daule and Babahoyo Rivers, is seen as the most prominent and evident feature of the sedimentation of the Guayas River (Soledispa, 2002). Its growth causes erosion along nearby banks (Dumont et al., 2007) and attracts birds that hinder take-off and landing of aircraft at the nearby international airport.

## 1.2 Major developments in the estuary

A number of factors have played a major role in the morphological development of the estuary and they will be assessed as part of this study:

1. *The construction of the Daule-Peripa Dam project*. The dam is located in the upper Daule river catchment, 186 km North from Guayaquil at the confluence of the Daule and Peripa rivers. Its construction was completed in 1988. The 78 m high and 250 m long earthfill dam creates an impoundment that covers 34,000 ha and stores over 6.0 $km^3$ (CELEC, 2013). The regulating effect of the dam is such that during the wet season the discharge is significantly lower than in the original situation, whereas during the dry months the situation is reversed. This has a direct effect on the sediment budget and transport rates of the Daule river, as well as a long-term effect on the river morphology. The situation prior to dam construction is referred to as Case 1 in the paper.

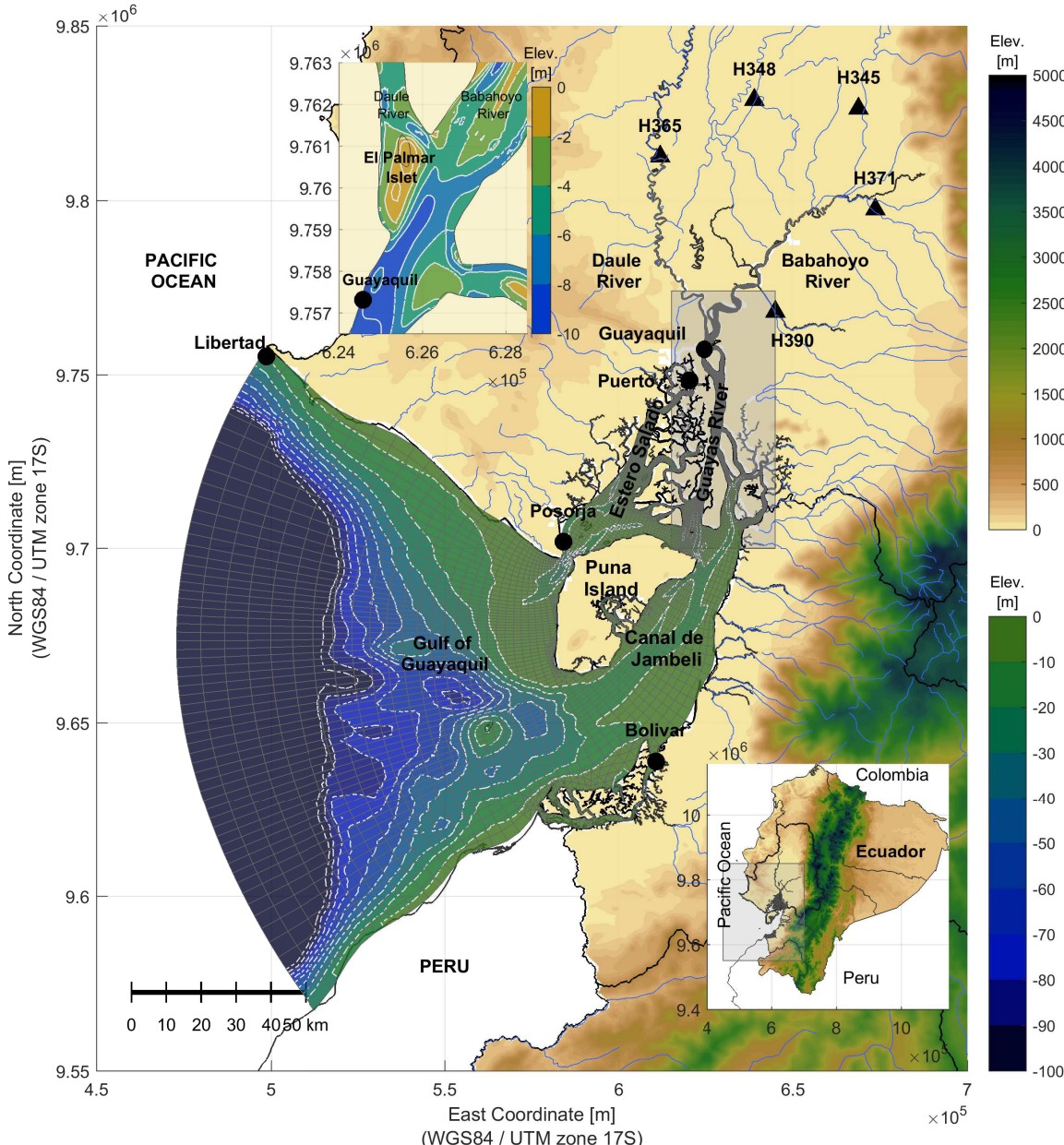

**Figure 1.** Map of the study area, covering the Gulf of Guayaquil, its estuary and the Guayas River. The map also includes the numerical grid of the Delft3D FM model. Tidal stations are indicated by circles, river gauging stations by triangles, while the area of interest is enclosed in the rectangle. The bathymetry is indicated as contour lines mapping depths every 10 m, with contour lines deeper than 100 m omitted from the map.

2. *Shrimp farming and mangrove deforestation*. Shrimp farming has intensified since 1970 and is the main cause which has led to mangrove deforestation and conversion into shrimp-raising lagoons (GADPG, 2013). Mangrove forests did not only help preventing erosion but also enhanced the sedimentation along the tidal flats. In addition, the presence of shrimp polders could lead to changes in tidal asymmetry, governing the net transport of the sediments. The effect of shrimp farms on tidal asymmetry has so far not been considered by previous studies. The situation prior to intense shrimp farming is referred to as Case 2 in the paper.

3. *The deforestation of the upper basin and changes in land use*. Deforestation accelerates the erosion processes, since the soil loses its natural cover, and becomes more exposed to the erosive effects of surface runoff, which in turn increases the sediment transport towards the rivers. The Guayas River basin covers 34,500 $\mathrm{km}^2$, and according to CAMAE (2013), it is estimated that on average 15 million tons of sediment are produced annually as a result of logging, changes in land use and landslides. The question is whether this increased sediment yield contributes to the sedimentation at the city of Guayaquil. This is referred to as Case 3 in the paper.

4. *The El Niño events*. The El Niño-Southern Oscillation (ENSO) phenomenon causes an influx of unusually warm surface water in the South-East Pacific Ocean. The resulting change in climate patterns may prompt torrential rainfalls on the coastal areas of Ecuador. The event typically peaks around December and lasts between nine months to two years. Severe floods are a direct consequence of El Niño. Moreover, especially in steep areas near the coast where clay soils are predominant, landslides may occur during heavy rainfall events. These effects do not only increase the discharge of rivers but also their sediment load. These conditions are referred to as Case 4 in the paper.

5. *Sea level rise*. As a consequence of global warming and climate change, a faster rise in mean sea level is expected in the following decades. Hallegatte et al. (2013) assessed the risks and economic losses due to flooding in 136 major coastal cities around the world. The risk assessment considered growth in future population, income and assets, as well as local subsidence and sea level rise scenarios between 20 and 40 $\mathrm{cm}$. According to the study, Guayaquil ranks fourth among the most vulnerable cities, with estimated economic average annual losses in the order of 3 billion USD. This is partly the result of relative sea level rise and partly to the limited capacity to adapt. In terms of morphological development, a rise in sea level implies an increase of the tidal prism and a consequent decrease of the intertidal areas. This situation is referred to as Case 5 in the paper.

## 2 Material and methods

### 2.1 Outline

A process-based numerical model was implemented in order to reproduce the morphological development of the Guayas River and the processes behind its evolution. The implementation of the model requires data of different nature that describe the boundary conditions and the geometry of the river. Among these data, the bed topography poses a particular problem, since

information in a suitable resolution is lacking. Therefore, prior the analysis, the derivation of a realistic initial bed topography was performed with the aid of a long-term morphodynamic simulation. The respective validation of hydrodynamics, sediment dynamics and morphodynamics followed. Finally, according to the major developments carried out in the estuary, a number of scenarios was defined. The analysis is focused on evaluating the effects on the sediment budget of each of the individual

scenarios, by comparison with a reference case that aims to mimic the actual situation.

## 2.2   Data collection

Data availability was one of the major challenges in carrying out this research. The following data types were collected and processed before being used as input to the numerical model: bed topography, water levels and currents, river discharges, salinity and sediment type.

The bed topography was obtained from the General Bathymetric Chart of the Oceans, (GEBCO, 2015). The data set is provided in a 30 arc-second resolution grid. This resolution is suitable for describing the outer part of the Gulf of Guayaquil. As bathymetry data were missing at a significant part of the inner estuary, a long-term morphological simulation was carried out starting from a flat bed in order to derive an input bed topography in equilibrium with the local hydrodynamic conditions in the estuary. The bed level is constructed following a common approach in 2D modelling of fluvial morphodynamics, also

applied by Van der Wegen and Roelvink (2012) to the Western Scheldt estuary in the Netherlands. Starting from a flat bed, in the area lacking topographic information, the model was forced until the formed channel-shoal patterns resembled the observed patterns in the areas where data were available. In particular, bed topography information was derived from a compilation of three surveys carried out by INOCAR in 2009. Additional surveys from March 2003 were available from USACE (2005). The latter information was used at locations not covered by the 2009 surveys.

Five tidal gauging stations spread over the Gulf and supported by the International Hydrographic Organization (IHO) were used for model calibration and validation: Libertad, Posorja, Puerto Bolívar, Puerto Marítimo and Guayaquil (Figure 1). Tidal current measurements are very scarce. Murray et al. (1976) determined the velocity field throughout the Gulf based on a field measuring campaign in October and November 1970. This information, which only included phase lags and velocities, is however rather local and schematized. Therefore, this information could only be used as general guideline for model validation.

River discharges were obtained from the hydrological yearbooks, published between 1984 and 2012 by the Ecuadorian National Institute of Meteorology and Hydrology, (INAMHI, 1984-2012). The discharges were converted into averaged monthly values, which were then used as input to the model. This allowed overcoming the issue with data gaps in the overall time series. In the Babahoyo River, the existing gauges are located at the 4 main tributaries (H345, H348, H371, H390 in Figure 1). For simplicity, the total contribution from the 4 tributaries was prescribed at one single location in the model. In the Daule River,

river discharges were obtained from one single gauge (H365). A total of three river discharge hydrographs were defined at the Daule River and two at the Babahoyo River, and used as input to drive the simulations of corresponding cases:

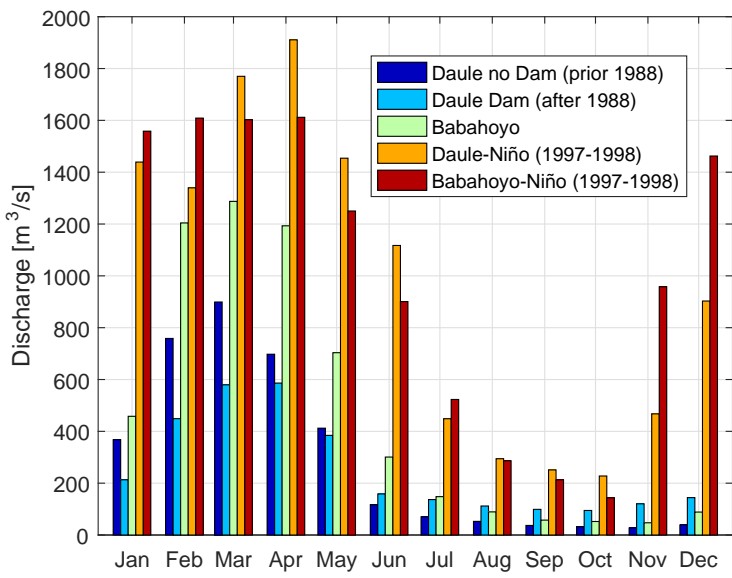

**Figure 2.** Monthly averaged discharge for the Babahoyo and Daule Rivers

– At the Daule River, a distinction was made between the average situation before and after construction of the Daule-Peripa Dam in 1988. In addition, the situation during one representative El Niño event from 1997-1998, and including the presence of the dam, was considered.

– At the Babahoyo River, only the average and the El Niño event conditions were simulated as no large dam is present at
5    this river branch.

Figure 2 presents the five hydrographs. The seasonality is evident, with a clear signal representing wet and dry seasons. Maximum discharges are observed between February and April, with the lowest discharges between August and October. The effect of the dam on the Daule River can be seen as a reduction in river discharge during the wet months January - May and an increase during the dry months June - December. Significantly larger discharges, compared to averaged years, can be observed
10   due to the El Niño events.

The estuary was assumed as fully mixed for its entire length and therefore the effects of density differences were disregarded. This is in agreement for example with Laraque et al. (2002) and Twilley et al. (2000).

According to Benites (1975), the mean sediment grain size in the estuary ranges between 200 and 400 μm. Fine sands and silts can be found at the confluence between the Daule and Babahoyo Rivers (USACE, 2005).

## 2.3 Numerical model

The process-based Delft3D FM (Flexible Mesh) was used in this study to simulate the coupled hydrodynamics, sediment transport and morphodynamic processes in the Guayas River (Kernkamp et al., 2011; Deltares, 2015). Delft3D FM solves the two- and three-dimensional shallow-water equations on an unstructured grid, based on the finite-volume method.

The model domain covers the entire Gulf of Guayaquil, from the edge of the continental shelf up to the upstream limit subject to tidal influence, approximately 50 km from the city of Guayaquil (Figure 1). Mosselman and Le (2016) argued that erroneous morphodynamic solutions may be obtained as a result of erroneous sediment input boundary conditions if boundaries are too close to the area of interest. In that regard, the 50 km distance between the area of interest and the boundaries is deemed sufficient to make the model domain suitable and keep the effect of disturbances to a minimum. The grid has a resolution

ranging between 80 m and 200 m in the area of interest and between 200 m up to 6,500 m in the outer zone of the Gulf.

At the seaward boundary, tidal conditions were derived from the global TOPEX/ Poseidon dataset (i.e., TPXO 2.0; Egbert and Erofeeva, 2002). Upstream mean monthly river discharges were derived from observations (Figure 2). Water levels at different tidal stations (i.e. Libertad, Posorja, Bolivar and Puerto) were used for model calibration (Figure 1). A list of the most relevant model parameters is presented in Table 1.

**Table 1.** Input model parameters

| Description | Parameter | Unit | Value |
|---|---|---|---|
| Smagorinsky constant | $c_{Smag}$ | - | 0.20 |
| Background eddy visc. | $v_{back}$ | [m$^2$.s$^{-1}$] | 10 |
| Manning friction coeff. | $n$ | [s.m$^{-1/3}$] | 0.0129 |
| Morphological factor | MorFac | - | 15 |
| Transport formulation | E-H | - | - |
| Median sediment diam. | $D_{50}$ | [µm] | 300 |
| Bed slope parameter | $A_{shld}$ | - | 0.4 |
| Bank erosion parameter | $\theta_{SD}$ | - | 0.5 |
| Spiral flow intensity | $E_{spir}$ | - | 1.00 |

The morphological boundary conditions were chosen such that the sediment input at the boundaries equals the transport capacity, and therefore the bed level at these locations experiences little to non-changes.

A uniform sediment fraction with a characteristic sediment size of 300 µm was defined for the entire model. For simplicity, and due to the very limited available data, the Engelund-Hansen sediment transport formulation was used. This formulation does not discriminate between suspended and bed-load transport. Therefore, the settling lag effect of fine suspended sediment

is not considered. However, spatial lag effects become important if the corresponding adaptation lengths are comparable with the size of the computational grid cells (Mosselman, 2005). Hence, the use of the Engelund-Hansen formulation can be justified as long as the cell sizes are larger than the required lengths for the processes of sediment suspension and deposition. In this

case, the grid resolution in the denser areas of the domain exhibits cell sizes in the order of 80 m, which could be assumed large enough to exceed the corresponding adaptation lengths linked to spatial lag effects of fine sediment.

Two main tuning parameters were used in the morphodynamic model to derive the starting bed topography for the simulation: the secondary flow ($E_{spir}$), controlling the intensity of the spiral flow, and the bed slope parameter ($A_{shld}$) to account for bed slope effect on the transport direction (Table 1).

In order to complete the simulation within an acceptable time frame, Roelvink (2006) suggested the introduction of a morphological acceleration factor (MorFac) to breach the difference between hydrodynamic and morphological time scales. For our study, a MorFac value equal to 15 was used (Table 1). In addition, input reduction techniques were used to schematize the hydrodynamic boundary conditions and couple them to the morphological acceleration factor. In particular, the tide was simplified to a mean "morphological tide" and the river discharges were scaled accordingly. The "morphological tide" concept is used to reproduce the same morphological changes as the complete tidal signal but using a shorter representative period (Lesser, 2009).

Monthly-averaged river discharges were associated to the time-varying tidal signal, by using the morphological acceleration factor (Guo et al., 2015). In particular, each monthly discharge was simulated over the duration of 2 morphological tide periods (i.e., approximately 2 days). Therefore, MorFac was selected in such a way that the morphological duration of those 2 tidal periods would match the duration of one month river discharge (i.e., MorFac ≈ 15).

Figure 3 illustrates the schematized tide and river boundary conditions for a year of morphological development. Each month was assumed to have equal duration, i.e., two morphological tide periods.

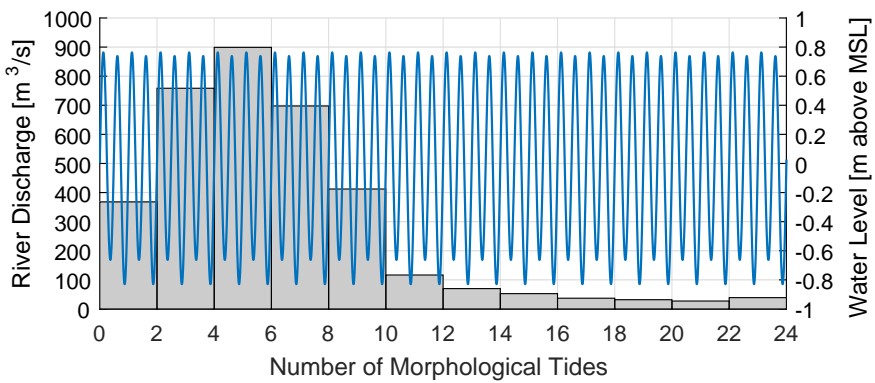

**Figure 3.** Schematization of the sea and river boundary conditions for one year of morphological development.

## 2.4 Model schematization of different cases

The following cases were simulated each for a period of 6 years (Table 2):

- *Case 0.* This reference case mimics the current state of the system. The discharge of the Daule River follows the hydrograph of the situation after the construction of the Daule-Peripa Dam (Figure 2). Additionally, the computational grid takes into account the width constriction along the Guayas River due to shrimp farm construction.

- *Case 1.* In this case, the hydrograph and sediment input prior to the construction of the Daule-Peripa dam were simulated (Figure 2).

- *Case 2.* In this case, the computational grid was modified in order to represent the situation prior to the reduction of intertidal areas by the development of shrimp farms. Additionally, the friction coefficient was increased to simulate the additional drag exerted by mangrove plants on the flow.

- *Case 3.* In this case, it was assumed that, due to deforestation, the sediment load is 1.20 times the transport capacity of the rivers at the boundaries. The sediment yield due to erosion in the upper basin of the Guayas River has not yet been formally estimated.

- *Case 4.* In this case, the effects of El Niño were assessed based on the associated higher discharges at the Daule and Babahoyo Rivers (Figure 2).

- *Case 5.* In this scenario, the effect of a 50 cm rise in mean sea level was assessed. This corresponds roughly to the local expected sea level rise at year 2100 under RCP (Representative Concentration Pathway) Scenario 4.5 or at year 2080 under RCP Scenario 8.5.

**Table 2.** List of cases simulated as part of this paper

| Case | Description |
|------|-------------|
| 0 | Actual situation |
| 1 | Prior to Daule-Peripa Dam project |
| 2 | Prior to shrimp farming and mangrove deforestation |
| 3 | Deforestation and changes in land use of the upper basin |
| 4 | El Niño |
| 5 | Sea level rise |

## 3  Results

### 3.1  Model calibration

The hydrodynamic model calibration was carried out by tuning the Manning friction coefficient to reproduce the observed water level signals at the different tidal stations. This was done with the aid of OpenDA, which is a set of tools for data-assimilation and calibration of numerical models. The tool runs simulations varying the Manning friction coefficient until an

optimum solution is obtained in relation to the observed water levels time series at the available tidal stations (Kurniawan et al., 2010). Model performances were assessed by means of the root mean square error (rmse) and the coefficient of determination ($R^2$). The best agreement between model results and measured data was obtained with a Manning friction coefficient of 0.0129 s.m$^{-1/3}$. This value agrees with what is often found in large rivers and estuaries, where bedform roughness is low if bedforms are elongated and mildly sloped. According to the formula ascribed to Strickler (1923) by Henderson (1966), the lower limit where flow resistance would be governed by grain roughness only yields $n = 0.034D_{50}^{1/6} = 0.009$ s.m$^{-1/3}$. Hence, the Manning value employed in the computations is above the minimum value for physically realistic hydraulic resistance. In other words, grain roughness is found to form about 70% of the total roughness and bedform roughness about 30%.

Table 3 summarizes the corresponding rmse and $R^2$ values at the different stations. The overall performance of the model was considered good given the limited available information.

In relation to morphodynamics, since bed level data were lacking for a significant part of the inner estuary, a long-term morphological simulation was performed in order to derive the missing information, as mentioned in subsection 2.2. In that regard, the topography for the entire Guayas River was initially set as a flat bed. A corresponding initial level of 6.00 m below mean sea level was determined based on the theory posed by Savenije (2006), that describes a general equilibrium state for alluvial estuaries in which the mean depth and the tidal amplitude remain constant along the estuary. The model then is run until some stable patterns are generated. The development of the estuary's depth (averaged over the domain) was used to assess the stability condition. In total, the morphodynamic simulation took about 200 years to reach the equilibrium, i.e., when the evolution of the estuary's depth remains constant. The obtained topography was contrasted against the few areas where information was available. It could then be verified that some characteristic observable features such as the formation of the "El Palmar Islet" were properly captured by the generated bed. As a final stage to validate the topography, the computed water levels were compared with those pertaining to measurements at the Guayaquil tidal station over a spring-neap tidal cycle. A similar model performance in relation to the results of the other tidal stations could be achieved. In addition, after performing a tidal analysis for both measured and modeled water levels, a generally good agreement could be verified for the amplitude and phases of the most energetic components. See Barrera Crespo (2016) for more detailed information.

The small water level bias in Table 3 indicates that derived depths of channels and shoals along the Guayas River agree with reality to a certain extent. This is supported since the tidal wave is not overly damped around low water as would be the case if the bed topography were too shallow.

**Table 3.** Statistical evaluation of model performances for tidal levels at different stations.

| Tidal Station | $R^2$ | rmse [m] | Wat. Lev. Bias [m] |
|---|---|---|---|
| Puerto | 0.907 | 0.359 | 0.0625 |
| Bolivar | 0.879 | 0.280 | -0.0101 |
| Posorja | 0.955 | 0.151 | 0.0173 |
| Libertad | 0.996 | 0.039 | -0.0152 |
| Guayaquil | 0.960 | 0.221 | -0.0151 |

## 3.2 Reference situation (Case 0)

The character of the tidal wave is estimated by analyzing the phase lag $\epsilon$ between high-water and high-water-slack. The left plot of Figure 4 maps $\epsilon$ throughout the Gulf of Guayaquil. Values close to 90 degrees indicate the wave has a propagating character. At the other end, a standing-wave-like character is dictated by values of $\epsilon$ close to 0 degrees. Both types of behavior are clearly

discernible by comparing the Estero Salado and the Guayas River. The former can be deemed as a tidal embayment since it lacks the input of fresh water, the latter on the other hand has a more straight shape and deeper topography that agrees with a more propagating behavior. In that regard, $\epsilon$ remains fairly constant along the Guayas River at approximately 35 degrees, i.e., the phase lag between high-water and high-water-slack is roughly 70 min.

The propagation of the tidal wave can be visualized by mapping the co-tidal lines throughout the Gulf. In the right plot of

Figure 4, co-tidal lines are mapped every 10 min while the color scale represents the amplitude of the $M_2 + S_2$ components, which can be used as proxy of the tidal amplitude at spring tide. The figure shows that it takes about 90 minutes for the tide to propagate from the Guayas River estuary mouth (northern from the Puna island) up to the Guayaquil tidal station. Moreover, the irregular spacing between the lines reveals that the wave celerity is not uniform and increases as the tide propagates up the Guayas River.

At the same time, the tide is amplified as it propagates up the estuary, due to convergence of the banks and as the bathymetry becomes shallower (e.g. Savenije, 2006). The maximum amplitude is found at the city of Guayaquil, at the confluence of the Daule and Babahoyo Rivers, where the tide is about twice as large as at the outer part of the Gulf. Along the Guayas River, the amplitude increases from about 1.6 m to about 1.9 m.

The distortion of the tidal signal plays a determinant role in the net sediment transport of tide dominated areas. Whether

a basin is flood- or ebb-dominant is determined by the magnitude of the maximum flood and ebb velocities respectively. In general, flood-dominance is enhanced by shallow estuaries, while ebb-dominance increases for deeper estuaries with large intertidal storage areas. There is a general feedback mechanism between tidal asymmetry and morphology. Tidal asymmetry is influenced by the morphology in the sense that the wave is distorted while it propagates (Van Rijn, 2011; Giardino et al., 2014). On the other hand, tidal asymmetry influences the morphological development through the residual sediment transport

(Hoitink et al., 2017).

A 24-hours record of water levels measured at Guayaquil gauging station during September 2015 is shown in Figure 5. The records show a longer falling period of approximately 7 hours compared to the rising period duration of 5 hours. This is an indication for the flood-dominant character of the estuary since shorter flood duration means that the maximum flood velocities are higher than the maximum ebb velocities. The difference gives rise to a tide-averaged residual current in the flood

propagation direction that induces a residual sediment transport in the same direction.

Ebb and flood durations mapped throughout the Gulf area are shown in Figure 6. The overall occurrence of shorter flood periods along the Guayas River proves the flood-dominant character of the estuary. Moreover, salinity profiles taken during the peak of the wet season along the Guayas River and at the confluence of the Daule and Babahoyo, prove that the estuary is well mixed (Twilley et al., 2000; Laraque et al., 2002). As a result, the impact of density driven flows caused by the discharges of

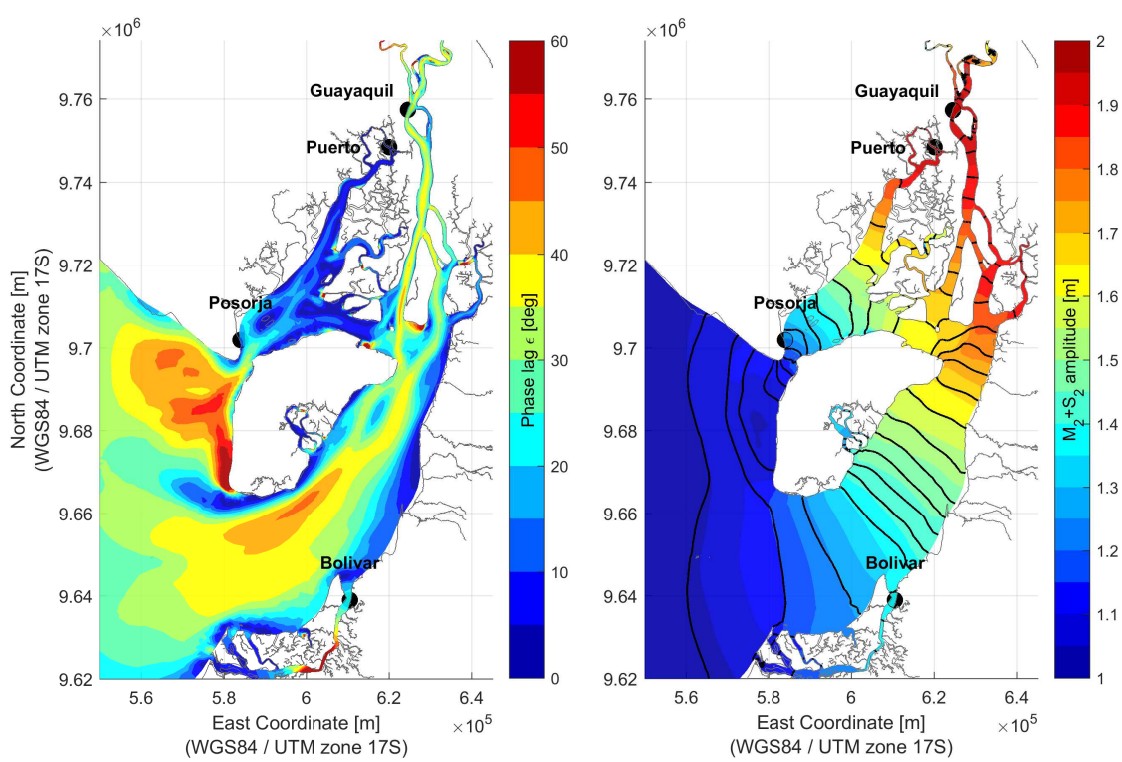

**Figure 4.** Phase lag between high-water and high-water-slack (left), tidal amplification and propagation throughout the Gulf of Guayaquil (co-tidal lines mapped every 10 min) (right)

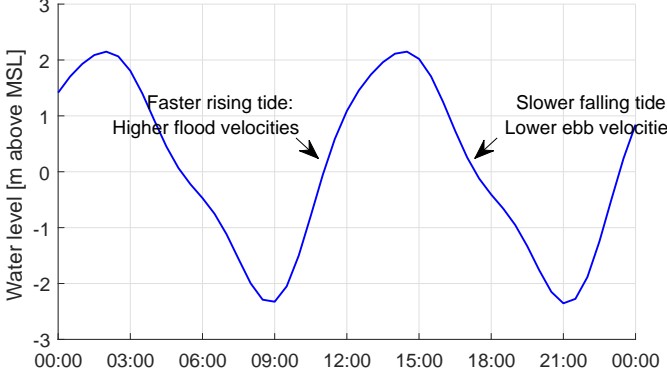

**Figure 5.** Tidal signal recorded at the Guayaquil gauging station.

the Daule and Babahoyo Rivers is mild on a yearly basis and the estuary can be regarded as tide dominated. This implies that the riverine input is likely to promote ebb-dominance only in the northern part of the estuary during large-discharge events.

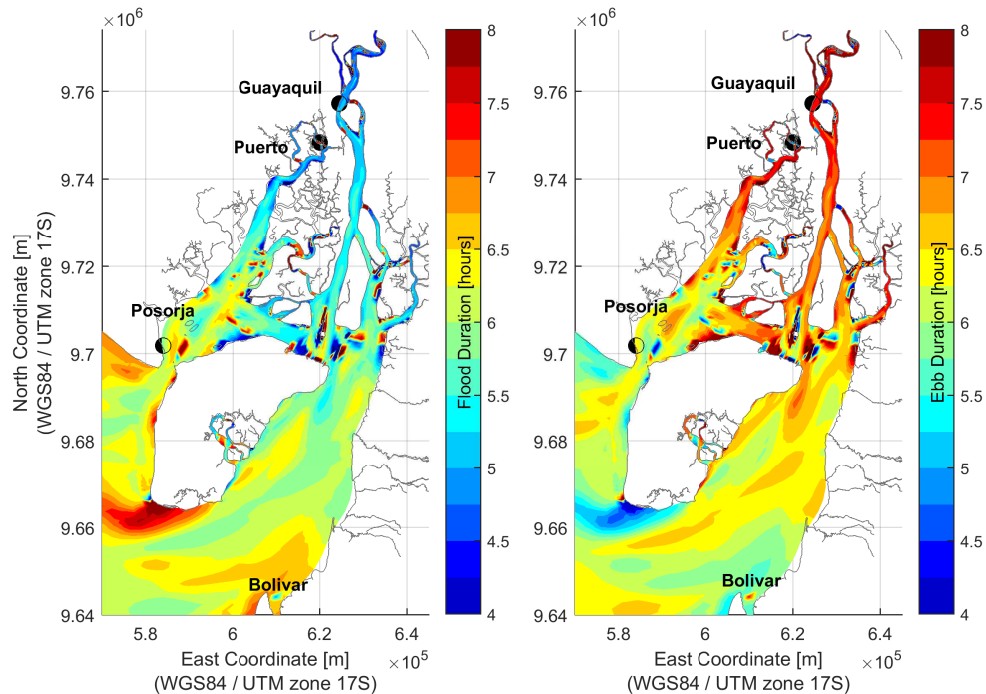

**Figure 6.** Duration of flood and ebb periods over a tidal cycle for the Gulf of Guayaquil

Besides the difference between flood and ebb durations, residual currents can also be the result of river discharges. As one moves up the estuary, ebb currents become larger until the point where there is no longer tidal influence and change in flow direction. Hence, the impact of changes in river discharges on the morphology are expected to be more evident in the northern part of the Guayas River. Residual currents were plotted in Figure 7 for both the dry and wet season. The plots were obtained by averaging the current velocity over the duration of a tidal cycle, when the river input is the weakest and strongest, during the dry and wet seasons respectively. For the dry season, the currents are directed upstream, in almost the entire length of the river, and they are stronger in the southern areas. During the wet season the situation is different, especially in the northern part of the river where the current direction reverts. Additionally, the magnitude of the upstream directed current is reduced.

These results confirm that, although the tide is the governing mechanism for the flood-dominant character of the system, the tidal river is also sensitive to the riverine input, particularly at the confluence of the Daule and Babahoyo Rivers . The changes in the residual current direction also imply a change in the residual sediment transport. These changes are especially noticeable in the zone near the confluence which can be considered as the most morphologically active zone of the Guayas River.

To determine the local sediment balance along the river, the domain was divided in 13 macro cells. The macro cells enclose areas within which the yearly residual sediment transport has approximately the same magnitude and direction (Jeuken and Wang, 2010). Erosion and sedimentation patterns at each cell as well as the sediment exchange between different cells were determined on a yearly basis as well as over the duration of the wet months (i.e. December to May) and dry months (i.e. June

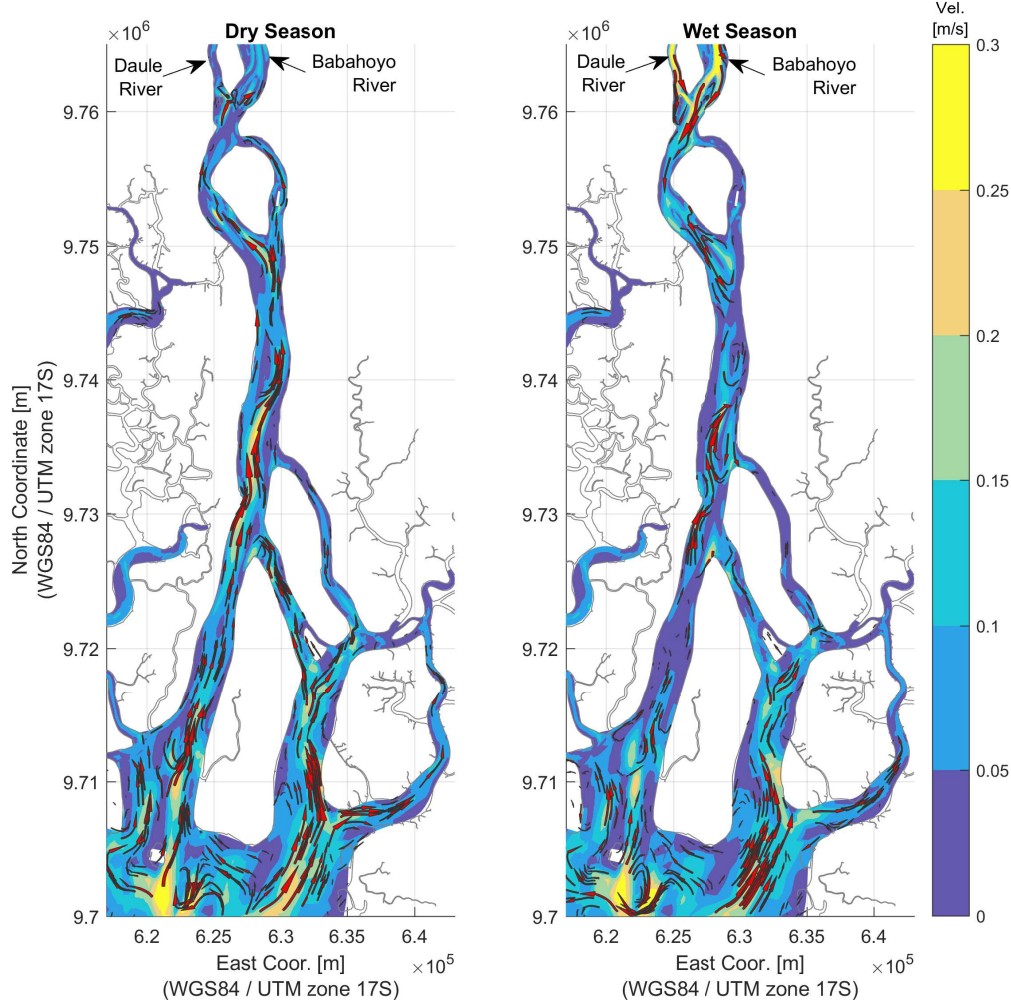

**Figure 7.** Tide-averaged residual current during the dry (left) and wet (right) seasons along the Guayas River.

to November), respectively. Figure 8 shows the computed sediment budget for the reference case, for the dry season, the wet season and the full year. During the dry season, the net transport is directed upstream. In contrast, during the wet season the transport along most of the river is directed downstream. This underlines the fact that the resulting magnitude and direction of the net transport is dependent on the relative riverine influence with respect to that of the tide. On a yearly basis, most of the Guayas remains flood-dominated with the exception of some of the northern cells. A general tendency towards sedimentation can be noticed for cells 5 and 6 (near Guayaquil) and cells 7, 10, 11, 12, and 13, which confirm the general observed trends in the area.

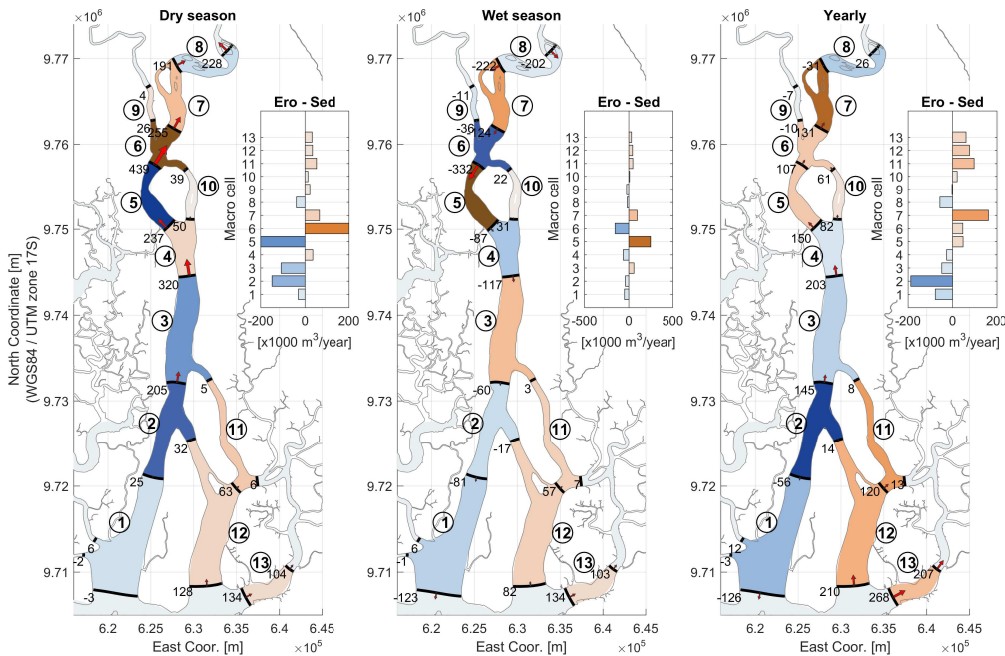

**Figure 8.** Sediment balance computed for the dry season (left figure), wet season (middle figure) and full year (right figure). Upstream sediment transport is considered positive whereas downstream transport is negative. The circle-enclosed numbers stand for the tag assigned to each of the macro cells. The numbers beside each transect quantify the actual sediment volume exchange between the cells (in x1000m$^3$/year). The overall sedimentation and erosion volumes for each macro cell are displayed as a bar plot.

### 3.3 Analysis of simulation cases

In this section, we analyze the results in terms of sediment transport and sediment balance for Cases 1 - 5. The difference between the cases relates to river discharges, mean sea level, geometry of the estuary and coarse sediment load transported by the rivers. The first three aspects have a direct influence on the tidal dynamics that determine ebb- or flood-dominance:
5 the effects of residual currents, convergence of energy and friction. The last aspect has a direct implication for the amount of sediment that enters the estuary. The influence of the different interventions is assessed by comparing each case with the reference case, which represents the actual situation (Case 0).

The figures show that, on a yearly basis, the river is importing sediment for most of the cases, except for the cases prior to the construction of shrimp polders (Figure 10) and during El Niño events (Figure 12).

10 The influence of the Daule-Peripa dam is mostly noticeable during the wet season (Figure 9). Prior to its construction, the unregulated discharges of the Daule River increased the effects of residual currents in downstream direction which, on a yearly basis, somewhat reduced the current flood-dominant character of the estuary.

Among the anthropogenic interventions, the construction of shrimp farms along the Guayas River had the largest effect on sedimentation (Figure 10). The situation prior to the construction of these polders reveals how these interventions have

led to a reverse in the direction of the residual sediment transport in most of the estuary, i.e., from ebb dominance to flood dominance. Intertidal flats, which were present prior the construction of the polders, provided additional friction which slowed down the upstream propagation of the tide. Moreover, when tidal flats were present, additional volumes of water could flow upstream during high tide (i.e. larger tidal prism). For continuity, the same volume of water had to flow downstream at ebb tide,

further increasing downstream flow velocities and therefore the tendency towards ebb dominance of the river. This is shown in Figure 10 by an overall increase in net erosion at most of the macro cells or a decrease in net sedimentation and sediment export towards downstream direction.

Figure 11 shows the results for enhanced sediment supply at the upstream river boundaries. These results are virtually the same as the results for the reference situation shown in Figure 7. Within the 6 simulated years, the sediment overloading

produced only some riverbed aggradation close to the upstream boundaries, indicating that it would take a very long time before increased bed-material load (sand) would produce any effect in the area of interest. Only increased washload (silt, clay) might have more immediate effects at Guayaquil, increasing sedimentation in stagnant water bodies in the area but having no effects on the main riverbed. Inspection of the development of stage-discharge relations at upstream gauge stations in the period 1990-2005 confirmed that even the upstream rivers do not show any signs of a sedimentation trend that could be related

to upstream deforestation or changes in land use. This suggests that the arbitrary chosen overloading by 20% is already an overestimate of the actual situation. Moreover, sand dredged from the Guayas River has a marine rather than a fluvial origin (Waumans, 2016).

The large river discharges associated with El Niño have a significant effect on promoting ebb-dominance (Figure 12). This is a direct consequence of the enhanced seaward residual transport that stems principally from the river discharges during the

wet season. The effect is more prominent in the upstream part where the riverine influence is larger. Sedimentation is promoted particularly in the middle sections 3 and 5, with erosion in the upper part, before the confluence of the Daule and Babahoyo Rivers.

Sea level rise basically increases the water depths throughout the estuary (Figure 13). This reduces bottom friction, which favours the upstream propagation of the high tide and hence promotes flood-dominance. Compared to the reference case, the

sediment balance exhibits a more flood-dominant character. As a result, sedimentation at the confluence of the Daule and Babahoyo Rivers (macro cells 6, 7 and 8) increases too.

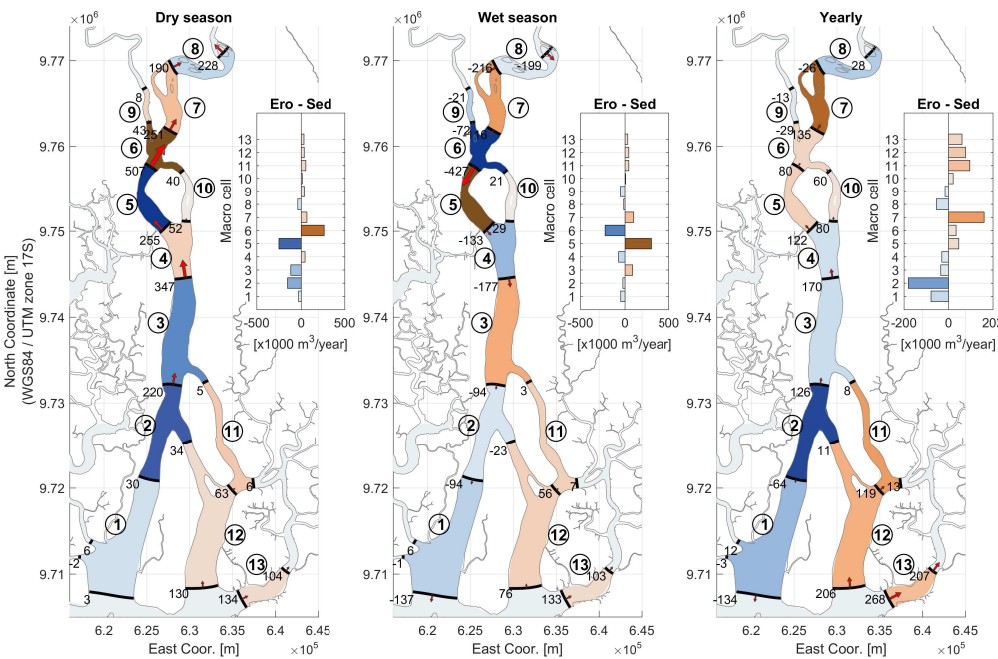

**Figure 9.** Sediment balance computed yearly and seasonally for the situation prior to the Daule-Peripa Dam project (Case 1).

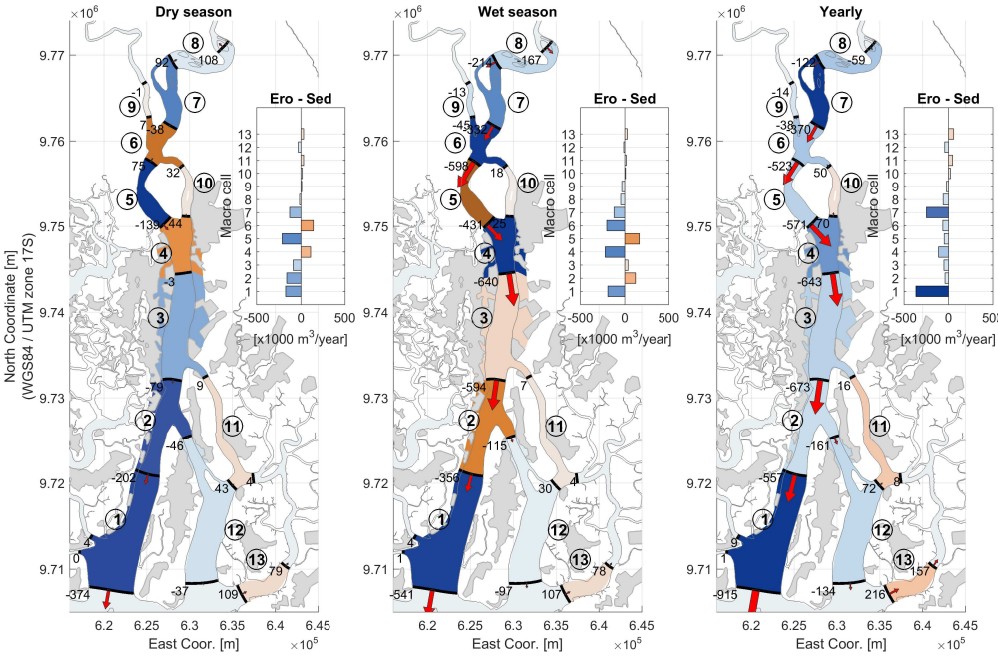

**Figure 10.** Sediment balance computed yearly and seasonally for the situation prior to shrimp farming and mangrove deforestation (Case 2). Areas covered by shrimp farms are depicted in lighter gray.

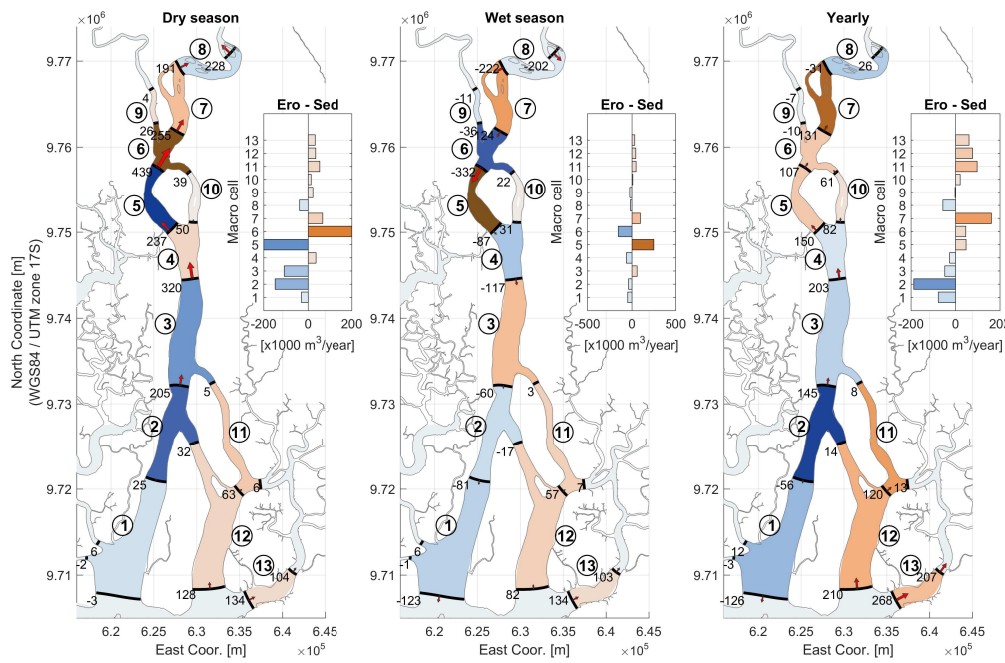

**Figure 11.** Sediment balance computed yearly and seasonally for deforestation and land use change in the upper basin (Case 3).

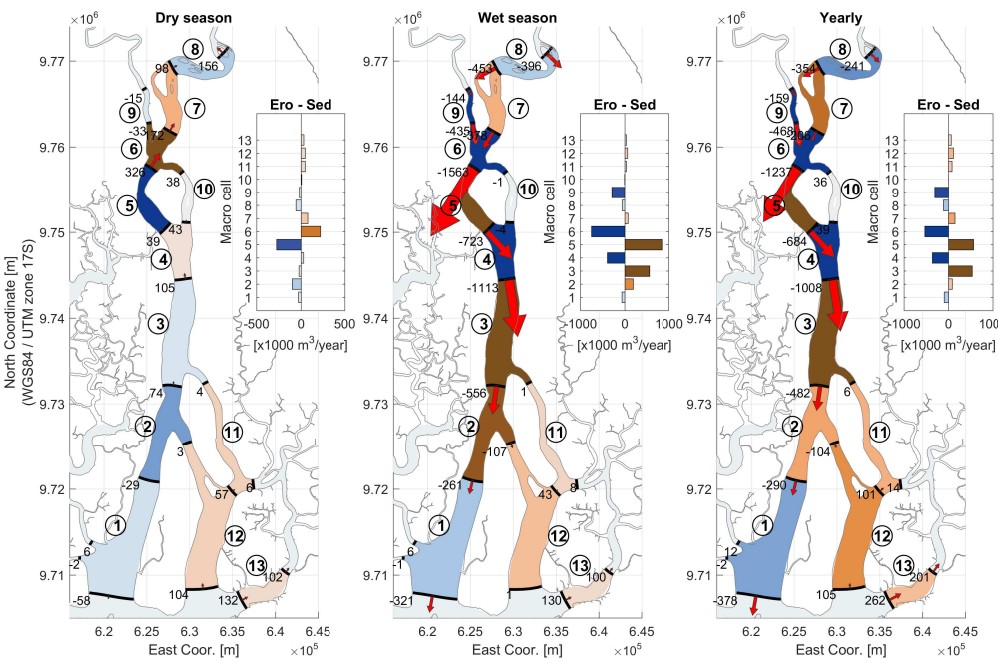

**Figure 12.** Sediment balance computed yearly and seasonally for El Niño (Case 4).

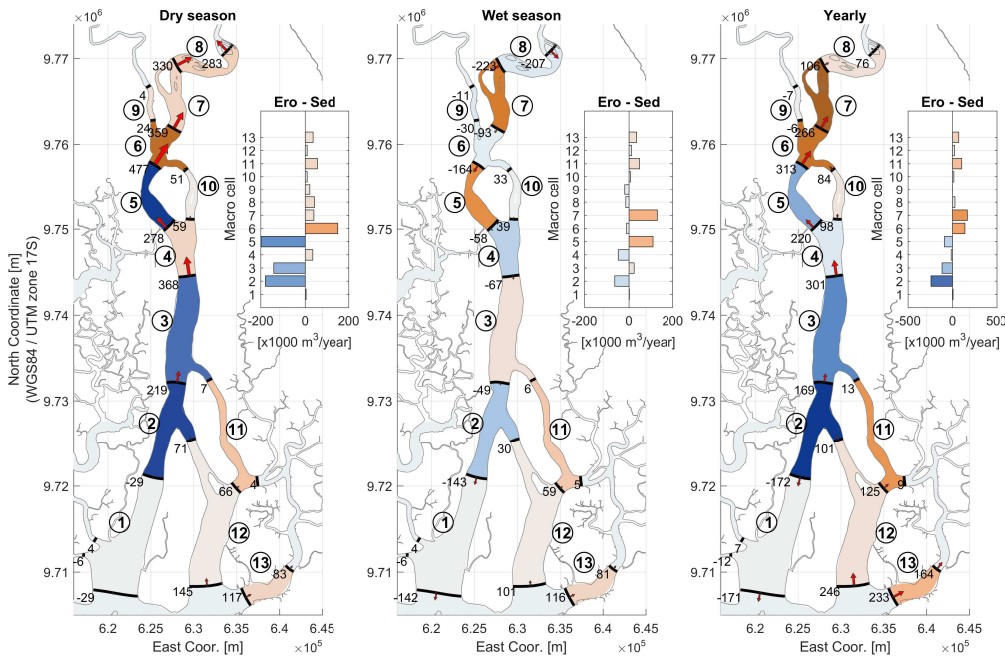

**Figure 13.** Sediment balance computed yearly and seasonally for sea level rise (Case 5).

# 4 Discussion

## 4.1 Data availability and uncertainties

A major factor of uncertainty relates to the availability of data for model input and calibration. The most prominent lack of data regards bed topography data along the river and its estuary, to be used for the reference situation as well as for assessing

5   morphological changes in time at the different macro-cells. For this reason, the initial bed topography was derived from a long-term morphological simulation, starting from a flat bed. The resulting bed levels were then validated with measured data at locations at which data were available.

Another major factor of uncertainty relates to the very limited information on the sediment type along the estuary and possible changes in sediment composition, which resulted from past interventions, natural events (e.g. El Niño events) or

10   climate change.

## 4.2 Sediment type and morphological evolution

Our analysis only considers sediment transport as total load, i.e. it does not discriminate between suspended and bed-load transport. Such a distinction could be important at small scales, but is not required at larger scales. Ignoring the difference between the two modes of transport is appropriate here, because the adaptation lengths for transport of the bed material (sand)

in suspension are smaller than five times the grid size, i.e. smaller than 5 x 80 m = 400 m (Galappatti and Vreugdenhil, 1985; Mosselman, 2005).

Contrary to the dependence of the transport of coarser bed material on flow strength, the transport of fine sediment depends on the sediment production of the upstream basin. This washload does quickly affect sediment concentrations at large distances downstream, but it is too fine for having any effect on the morphological evolution of the Guayas River. Yet it could lead to deposition in relatively stagnant flow, such as in storm sewer outfalls, and in estuaries further downstream where fine sediment particles may flocculate in contact with higher salinity. We did not study sediment processes in the estuaries, but limited our study to the morphological development of the Guayas River.

The local erroneous attribution of sedimentation in the Guayas River to upstream deforestation fits in a wider pattern of perceptions and believes regarding the root causes of sedimentation in different rivers around the globe. Examples include sedimentation by embankments, barrages and water abstraction (e.g. Indus River in Pakistan, Gaurav et al. (2011)), backwater effects at low discharges (e.g. Magdalena River at Barranquilla in Colombia, Garay (2015)), backwater effects due to rapid sea level rise (e.g. rivers discharges into the Caspian Sea, Sloff (2014)) or unstable bifurcations and avulsions in deltas and megafans (e.g. Taquarí River in Brazil, Makaske et al. (2012)). Notwithstanding potential local benefits, reforestation often will not bring the benefits presumed from mitigation of sedimentation far downstream.

## 4.3 Long-term morphological changes

In the study, the effect of past interventions, natural events or climate change on the sediment transport and erosion/depositions patterns along the estuary have been simulated through a scenario analysis. In particular, the computed sediment budget are representative of the direct (initial) effects which will result from these interventions (i.e. morphostatic approach). It is important to realize that these changes in reality will result into morphological adjustments of the entire estuary in the longer term.

## 5 Conclusions

We analysed the sediment transport and morphological development of the Guayas River as a result of past interventions, natural events and climate change by means of a process-based numerical model. The picture arises that the sediment balance around Guayaquil is governed by sand import from downstream, owing to the flood-dominant character of the tide, and periodic flushing out of this sand by river floods. This balance has been disturbed in two ways. First, reclamation of intertidal areas for shrimp farming has made the tide more flood-dominant. This has increased the sand import from downstream. Second, flow regulation at the Daule-Peripa Dam has decreased the discharges during floods, thus decreasing the periodic flushing. The combined effect has produced stronger river sedimentation at the city of Guayaquil. Contrary to local perception, upstream deforestation has no effect on this. At most, it might have enhanced the deposition of fine sediment in connected stagnant water bodies, such as storm sewer outfalls, and in the estuaries further downstream. This deposition, however, was outside the

scope of our study. El Niño events play an important role in producing river floods that flush out sediment. Sea level rise would increase the sediment import into the river as it would make the tide even more flood-dominant.

Measures to mitigate sedimentation at Guayaquil should be directed towards increasing the effect of the large river discharges and minimizing the effect of the distortion of the tidal wave, by modifying the shape of the estuary. This could be achieved, for instance, by managing the operation of Daule-Peripa dam and by recovering some of the decimated intertidal flats. A third alternative could be the dredging of the most vulnerable areas. A definitive solution would involve a compromise between the cost of dredging and the decreased profit from hydro-power generation and aquaculture practices. Further research is needed to define and evaluate the extent, viability, and socio-economic and environmental impacts related to each of the potential measures to be implemented.

*Acknowledgements.* Otto de Keizer arranged the conditions and the contacts in Guayaquil to carry out this study. We are grateful to EMA-PAG, INOCAR, INTECSA-INARSA and Jacinto Rivero Solórzano for sharing data and providing information. We thank Erik Waumans of Van Oord for confirming that the material dredged in the Guayas River is sand of marine origin.

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
