# Peer review of "Sediment budget analysis of the Guayas River using a process-based model"

_Hydrology and Earth System Sciences, 2018_

## Referee Comment (RC1) · M. van der Wegen (Referee) · 16 Nov 2018

Dear Authors;

Last week I read with great interest your work on the sediment budget analysis of the Guaya River system. You applied a process-based model to derive a sediment budget in a data scarce environment. The modeling steps you take are somewhat crude. Still I like the approach since it make use of the capacity of process-based models to understand and describe system dynamics, even without much data.

Pls consider my minor comments in the attached document; My major concerns are

summarized below:

1) Your methodology is as follows . 1-derive a 'realistic' bathymetry starting from a flat bed. 2- validate hydrodynamics-sediment dynamics and morphodynamics. 3- evaluate model results and scenarios. Update your methodology section so that it reads like this (especially wrt the morphodynamic validation) 2) At page 17 , line 7-8 you state that the resulting bed levels are validated against data. I think this is crucial and should be mentioned much earlier, eg in chapter 3. Also answer questions like; How realistic is the generated bed? Did you start from a flat bed ? How did you determine the initial bed level? What is the impact of a different initial bed level, eg on the water level bias? Was the generated bed in equilibrium? How long did that take? 3) You relate the outcome of the scenarios to tidal asymmetry, but why not show that in terms of ebb/flood duration or changing Stokes' return flow (eg see Van der Wegen et al 2008 and refs therein)? 4) As you admitted the EH transport is not really suitable for finer sediments. Even more, density driven (salt-fresh) flows may have considerable impact as well even on both fine and coarse sediments (see Effects of Density‐Driven Flows on the Long‐Term Morphodynamic Evolution of Funnel‐Shaped Estuaries Maitane Olabarrieta W. Rockwell Geyer Giovanni Coco Carl T. Friedrichs Zhendong Cao doi:10.1029/2017JF004527). That make your conclusions vulnerable; pls discuss 4) Because of the many crude assumptions (still, necessarily taken), related to , waves, sediment size, tidal movement density flows etc. you should present the results more as indicative in the abstract and conclusions.

I hope you consider the above comments an encouragement to adjust the work. I am looking forward to an update

Mick van der Wegen

Please also note the supplement to this comment:
https://www.hydrol-earth-syst-sci-discuss.net/hess-2018-467/hess-2018-467-RC1-supplement.pdf

[Figure]

**Supplement:**

[revised manuscript text omitted]

---

## Referee Comment (RC2) · D. Zagar (Referee) · 21 Nov 2018

Dear Authors,

I read the manuscript Sediment budget analysis of the Guayas River using a process-based model and I found it very interesting. The results of the performed research enabled a comparison of the contribution of various anthropogenic and natural impacts on sediment transport in the Guayas River. Some of the findings and conclusions presented in this article could undoubtedly help to further mitigation measures preventing excessive sedimentation along the river.

[Figure]

My major concerns are the following:

1. The accuracy of the initial riverbed topography modelling: nowhere in the text the agreement with (scarce, definitely, but not entirely non-existent!) measurements are mentioned. Even the qualitative agreement (of phenomena – sedimentation/erosion) is questionable with simulations using a poorly matched riverbed. Quantification of sedimentation/erosion is of course even more questionable. A comparison (where available) would help to increase the scientific and practical value of the results.

2. Very coarse river discharge data. The monthly averaged discharge presented in figure 2 lies within a relatively large range of discharges between the months. What is the daily discharge dynamics? At least a reference to a (more or less detailed) hydrology-study of the river(s) under consideration would be very helpful. Without at least the range of (minimum/maximum) discharge or the variance of discharge within a month all short-term events are excluded from simulations. Possible extreme events are never mentioned in the text (is there none?). Moreover, the non-linear dependence of sediment transport on discharge increases the significance of short-term high-discharge events. An out-of-season extreme discharge could significantly change the quantities of downstream sediment transport as well as the conclusions given by the authors. These two questions are, in my opinion, crucial for justification of the conclusions. Nonetheless, I would like to encourage the authors to improve the manuscript and to explain the accuracy of the applied procedures and the simulated processes. Other minor remarks and suggestions can be found in the text, uploaded as a supplement.

I will be pleased to re-read the improved manuscript!

Dušan Žagar

Please also note the supplement to this comment:
https://www.hydrol-earth-syst-sci-discuss.net/hess-2018-467/hess-2018-467-RC2-supplement.pdf

[Figure]

[Figure]

**Supplement:**

[revised manuscript text omitted]

---

## Referee Comment (RC3) · Anonymous Referee #3 · 4 Jan 2019

I really enjoyed reading this article because the work done is an excellent contribution for the understanding of the processes that govern sediment transport dynamics in the Guayas River in order to assess the trade-offs for urban risk management of coastal cities as Guayaquil.

After a careful review of the article, I consider that some considerations should be included for the scenario definition. In scenario 1, I would suggest to include a more realistic representation of the spatial distribution of tidal flats in the inner estuary previous to the construction of Daule-Peripa dam. The scenario 4 did not consider the increment on mean sea level (MSL) due to thermal expansion because of El Niño conditions. For

the 1997-1998 event, Zambrano et al. (2000) reported a maximum increase of 42 cm on MSL at La Libertad, in Ecuador. Including this effect for the simulation of scenario 4 could provide interesting insights on the net effect of a strong El Niño event on the sediment transport dynamics in the Guayas River.

───────────────────────

---

## Author Comment (AC1) · 31 Jan 2019

**Response to the comments made by Reviewer 1: Mick van der Wegen**

We would like to thank the Reviewer for the time spent revising the manuscript and the detailed comments provided. Please find below responses to the each of your remarks about the manuscript in a comment-by-comment basis.

**R1-Comment 1: At page 17, line 7-8 you state that the resulting bed levels are validated against data. I think this is crucial and should be mentioned much earlier, e.g. in chapter 3. Also answer questions like; How realistic is the generated bed? Did you start from a flat bed? How did you determine the initial bed level? What is the impact of a different initial bed level, e.g. on the water level bias? Was the generated bed in equilibrium? How long did that take?**

We thank the Reviewer for the suggestion. We agree and will update the manuscript including a description of the procedure to derive the initial bathymetry. Nevertheless, a detailed explanation on the derivation and validation of the initial bathymetry implies an extensive description which would occupy much of the journal space. In that regard, please refer to the master thesis by Barrera Crespo (2016). Delft3D Flexible Mesh modelling of the Guayas River and Estuary system in Ecuador. Delft University of Technology, National University of Singapore, http://resolver.tudelft.nl/uuid:c8a4c2f1-208b-4332-a17f-8afb28ec71e6. The thesis presents a complete description of the procedure, which is similar to what is proposed by Van der Wegen and Roelvink (2012) for the Western Scheldt estuary in the Netherlands, and all of the above questions are answered as follows:

- How did you determine the initial bed level?
  An analytical model of the estuary was implemented based on the theory posed by Savenije (2006) about alluvial estuaries. The theory describes a general equilibrium state for alluvial estuaries in which the tidal amplitude remains constant as it propagates throughout. With this model then it is possible to assess the equilibrium condition and the subsequent mean water depth for the entire Guayas River Estuary.

- Did you start from a flat bed?
  The Delft3D morphodynamic model initial condition renders a flat bed in the area lacking topographic information. The bed level was set according to the mean water depth estimated with the analytical model.

- Was the generated bed in equilibrium? How long did that take?
  The morphodynamic simulation was run until the equilibrium condition is reached. This was determined by monitoring the evolution of the mean estuary's depth and sedimentation and erosion volumes in the area lacking topographic information. In total it took about 200 years of morphodynamic simulation to reach the equilibrium.

- How realistic is the generated bed?
  The available topographic data is scarce, only a few contour lines spaced vertically every 5 m were available in some areas of the estuary. These contours cover elevations that range between -2 and -20 m in relation to MSL. The latter information was complemented with available satellite images in order to discern characteristic morphological features like shoals and deep channels. This was

then contrasted with the topography derived by the long-term simulation. After a close visual inspection, it could be verified that most of the morphological features were captured properly in the northern and middle part of the estuary. The southern part presents larger discrepancies that could be ascribed to the presence of non-erodible layers in the proximities of the Puna Island. The final stage to validate the derived topography is to contrast the computed water levels with those pertaining to measurements at the Guayaquil tidal station over a spring-neap tidal cycle. A good correlation and a small mean square error between both measured and modelled water levels were obtained. In addition, after performing a tidal analysis for both signals, a generally good agreement could be verified for the amplitude and phases of the most energetic components.

**R1-Comment 2: You relate the outcome of the scenarios to tidal asymmetry, but why not show that in terms of ebb/flood duration or changing Stokes' return flow (e.g. see Van der Wegen et al 2008 and refs therein)?**

We thank the Reviewer for raising to our attention the analysis regarding tidal asymmetry in terms of ebb/flood durations. Tidal asymmetry is a concept that encompasses the asymmetry of the vertical and horizontal tide. The asymmetry of the vertical tide is linked to the difference between flood and ebb periods. As a consequence, flood-dominance renders a rising period shorter than a falling period. In the other hand, the asymmetry of the horizontal tide relates to 2 aspects: the difference between ebb and flood velocities, and the duration of the slacks. In relation to the first aspect, flood-dominance happens when the maximum flood velocity is larger than the maximum ebb velocity. For the second aspect, flood-dominance occurs when the duration of high-water slack is longer than the low-water slack.
In figures 5 and 6 we show exactly the flood-dominant behavior of the Guayas estuary as a function of the vertical and horizontal tides respectively. However, we will complement the manuscript with an analysis of the duration of the slack water throughout the estuary.

**R1-Comment 3: As you admitted the EH transport is not really suitable for finer sediments. Even more, density driven (salt-fresh) flows may have considerable impact as well even on both fine and coarse sediments (see Effects of Density Driven Flows on the Long-Term Morphodynamic Evolution of Funnel-Shaped Estuaries Maitane Olabarrieta W. Rockwell Geyer Giovanni Coco Carl T. Friedrichs Zhendong Cao doi:10.1029/2017JF004527). That make your conclusions vulnerable; pls discuss. Because of the many crude assumptions (still, necessarily taken), related to, waves, sediment size, tidal movement density flows etc. you should present the results more as indicative in the abstract and conclusions.**

Again, we thank the Reviewer for raising this to our attention. According to the analysis presented in Barrera Crespo (2016), the importance of the river discharges is minor in relation to the tide. As a result, the Guayas river is deemed tide dominated and becomes partially mixed during the peak of the wet season, and well mixed the rest of the year, especially during the dry season. Based on this, the occurrence and impact of density driven flows is mild in a yearly basis. Measurements of actual salinity profiles taken along the Guayas estuary support these findings e.g. Twilley et al. (2000), and Laraque et al. (2002).
The distribution of sediment size throughout the estuary presented by Benites (1975) suggests a relatively coarse sediment size of 0.30 mm, this was confirmed by a personal communication from a dredger (Erik Waumans of Van Oord). Moreover, settling lag effects of suspended sediments become important if the corresponding adaptation lengths are comparable with the size of the grid cells, Mosselman (2005). In this

case, the grid resolution in the denser areas of the model renders cell sizes that are much larger than the adaptation lengths of fine suspended sediments. This was the basis to justify the usage of the Engelund-Hansen formulation.

For completeness, we will include in the manuscript the discussion of density effects and the appropriateness of the Engelund-Hansen formula.

**R1-Comment 4: Minor comments to the manuscript.**

We appreciate the thorough revision made by the reviewer to the overall redaction of the manuscript, we will abide by the suggestions regarding the rephrasing of some sentences. We will also include the visualization of the El Palmar islet within Figure 1 as shown below:

[Figure]

To support the discussion about the use of the morphological factor in tide-river environments, and the use of the OpenDA tool for calibration purposes we will include the following references:

- Guo, L., van der Wegen, M., Roelvink, D., & He, Q. (2015). Exploration of the impact of seasonal river discharge variations on long-term estuarine morphodynamic behavior. Coastal Engineering, 95, 105-116.

- Kurniawan, A., Ooi S. K., H. Gerritsen and D.J. Twigt. Calibrating the regional tidal prediction of the Singapore Regional Model using OpenDA

**References**

Barrera Crespo, P. D. (2016). "Delft3D Flexible Mesh modelling of the Guayas River and Estuary system in Ecuador." Delft University of Technology, National University of Singapore, Delft. (http://resolver.tudelft.nl/uuid:c8a4c2f1-208b-4332-a17f-8afb28ec71e6).

Benites, S. (1975). "Morfología y sedimentos de la Plataforma Continental del Golfo de Guayaquil." *Tesis ESPOL* 112.

Laraque, A., Cerón, C., Magat, P., and Pombosa, R. (2002). "Informe del primer estudio del impacto de la marea sobre el estuario del Río Guayas.".

Mosselman, E. (2005). "Basic equations for sediment transport in CFD for fluvial morphodynamics." *Computational fluid dynamics: applications in environmental hydraulics* 71-89.

Savenije, H. H. G. (2006). Salinity and tides in alluvial estuaries. Elsevier.

Twilley, R. R., Cárdenas, W., Rivera-Monroy, V. H., Espinoza, J., Suescum, R., Armijos, M. M., and Solórzano, L. (2000). "17 The Gulf of Guayaquil and the Guayas River Estuary, Ecuador." *Coastal Marine Ecosystems of Latin America* 144:245.

Van der Wegen, M. and Roelvink, J. A. (2012). "Reproduction of estuarine bathymetry by means of a process-based model: Western Scheldt case study, the Netherlands." *Geomorphology* 179:152-167.

---

## Author Comment (AC2) · 31 Jan 2019

**Response to the comments made by Reviewer 2: Dušan Žagar**

We thank the Reviewer for the meticulous revision of the manuscript. Please find below responses to the each of your remarks in a comment-by-comment basis.

**R2-Comment 1: The accuracy of the initial riverbed topography modelling: nowhere in the text the agreement with (scarce, definitely, but not entirely non-existent!) measurements are mentioned. Even the qualitative agreement (of phenomena – sedimentation/erosion) is questionable with simulations using a poorly matched riverbed. Quantification of sedimentation/erosion is of course even more questionable. A comparison (where available) would help to increase the scientific and practical value of the results.**

We thank the Reviewer for the constructive criticism. This comment is similar to what was pointed out by Reviewer 1, so we kindly refer to the respective response for a complete explanation. However, as stated in the response, we will include a description of the procedure to derive and validate the bathymetry, as well as a comparison with actual measurement where the available information so allows it. Due to the scarce available topographic data a more rigorous comparison is not possible.
For a complete and detailed description of the on the derivation and validation of the initial bathymetry please refer to the master thesis by Barrera Crespo (2016). Delft3D Flexible Mesh modelling of the Guayas River and Estuary system in Ecuador. Delft University of Technology, National University of Singapore, http://resolver.tudelft.nl/uuid:c8a4c2f1-208b-4332-a17f-8afb28ec71e6

**R2-Comment 2: Very coarse river discharge data. The monthly averaged discharge presented in figure 2 lies within a relatively large range of discharges between the months. What is the daily discharge dynamics? At least a reference to a (more or less detailed) hydrology study of the river(s) under consideration would be very helpful. Without at least the range of (minimum/maximum) discharge or the variance of discharge within a month all short-term events are excluded from simulations. Possible extreme events are never mentioned in the text (is there none?).**

We thank the Reviewer for pointing this out. Nevertheless, we are not aware of any recent hydrological study that would demonstrate the relevance of including such daily variations. In the manuscript we present mid-term effects in a yearly basis, within this context short-terms events are averaged out and don not affect the overall sediment balance. Moreover, the discharges are imposed as boundary conditions for the Daule and Babahoyo rivers. Each of these boundaries are located about 50 km landward from the confluence of both rivers, so there is no immediate or direct influence of the imposed discharges in the area of interest. Once the at the confluence, the influence of the tide dominates the flow dynamics.

**R2-Comment 3: Minor comments to the manuscript.**

We appreciate the extensive revision made by the reviewer to the wording of the manuscript. Attending the suggestions we will include the visualization of the El Palmar islet in Figure 1. Additionally, regarding the examples presented subsection 4.2, we derived knowledge based on our involvement in projects with direct contact with data and local experts. In that regard, we will add the following references:

- Garay Bohórquez, C. (2015), Personal communication. CorMagdalena, Colombia.

- Gaurav, K., R. Sinha & P.K. Panda (2011), The Indus flood of 2010 in Pakistan: a perspective analysis using remote sensing data. Natural Hazards, Journal of the International Society for the Prevention and Mitigation of Natural Hazards, ISSN 0921-030X, DOI 10.1007/s11069-011-9869-6.
- Sloff, C.J. (2014), Personal communication on sedimentation in the lower Kura River in Azerbaijan. River morphology expert involved in Flood Prevention Program Azerbaijan.

Finally, in the same subsection 4.2, we will also change the sentence "Examples include sedimentation by water abstraction (e.g. Indus River in Pakistan)" into "Examples include sedimentation by embankments, barrages and water abstraction (e.g. Indus River in Pakistan)".

**References**

Barrera Crespo, P. D. (2016). "Delft3D Flexible Mesh modelling of the Guayas River and Estuary system in Ecuador." Delft University of Technology, National University of Singapore, Delft. (http://resolver.tudelft.nl/uuid:c8a4c2f1-208b-4332-a17f-8afb28ec71e6).

---

## Author Comment (AC3) · 31 Jan 2019

**Response to the comments made by Reviewer 3: Anonymous**

We thank the Reviewer for your remarks about the scenario definitions. Please find below responses to the each of your remarks in a comment-by-comment basis.

**R3-Comment 1: In scenario 1, I would suggest to include a more realistic representation of the spatial distribution of tidal flats in the inner estuary previous to the construction of Daule-Peripa dam.**

What we try to assess in scenario 1 is the influence of the regulated discharges in the Daule river due to the construction of the Daule Peripa dam in the upper basin. Moreover, we address the representation of the decimated tidal flats before the construction of shrimp farms in case 2.

**R3-Comment 2: The scenario 4 did not consider the increment on mean sea level (MSL) due to thermal expansion because of El Niño conditions.**

In the manuscript we treat increased riverine input due to El Niño and MSL rise separately. So, we define case 4 based on the increased riverine input, and case 5 based solely on MSL rise. By comparing figures 11 and 12 it can be seen, on a yearly basis, that sediment transport rates due to increased discharges are much larger than for the sea level rise case. In that regard, the results of a combined case are not going to differ much from those of case 4.

---

## Referee Report (RR1)

Comments on the revised version 2019/05/15

The manuscript Sediment budget analysis of the Guayas River using a process-based model by P.D Barrera Crespo et al. has been significantly improved. Considering the low amount of available input data, the authors clarified their procedures and provided satisfactory replies to the concerns and remarks reported previously.

I am suggesting the revised version of the manuscript to be published in the HESS journal with a single technical correction suggested:

P9-line15: please replace "somewhat constant" with a more suitable formulation; something (tidal amplitude) can be either constant or variable.

Dušan Žagar

---

## Author Response (AR2)

**Response to the comments after the revision made by the editor: Matjaž Mikoš**

We would like to thank the Editor for the time spent revising the manuscript. Please find below responses to the each of your remarks about the manuscript in a comment-by-comment basis.

**Editor-Comment 1: p. 2, line 10 - the number of inhabitants/population in the City of Guayaquil would be nice to read.**

According to the editor suggestion we modified the text in p.2 line 10 as follows:

Guayaquil, "La Perla del Pacífico" (The Pearl of the Pacific), is the most populated city (2,644,891 inhabitants) (INEC,2017) and the industrial and commercial capital of Ecuador.

**Editor -Comment 2: p. 2, line 25 - could you add the height of the dam? Any discussion on the dam operation on flushing (yes/no) and the siltation rate of the reservoir would add to the paper.**

According to the editor suggestion we modified the text in p.2 line 26 as follows:

The 78 m high and 250 m long earthfill dam creates an impoundment that covers 34,000 ha and stores over 6.0 $km^3$ (CELEC, 2013).

**Editor-Comment 3: p. 4, line 9 - could you also give specific sediment production in t/km2 or kg/ha, we do not know exactly to what area we should use the average of 15 million tons of sediment per year.**

There is no exact estimation of the sediment production in the upper basin, so instead of value in $t/km^2$ we added the extension area of the Guayas River basin for reference. According to the editor suggestion we modified the text in p.4 line 9 as follows:

The Guayas River basin covers 34,500 $km^2$, and according to CAMAE (2013), it is estimated that on average 15 million tons of sediment are produced annually as a result of logging, changes in land use and landslides.

**Editor-Comment 4: p. 9, line 8 - could you comment on the value of the Manning friction coefficient of 0.0129 (low value) how good this values corresponds to other similar cases in the world?**

According to the editor suggestion we added the following the text in p.10 line 4:

This value agrees with what is often found in large rivers and estuaries, where bedform roughness is low if bedforms are elongated and mildly sloped. According to the formula ascribed to Strickler (1923) by Henderson (1966), the lower limit where flow resistance would be governed by grain roughness only yields $n=0.034\ D_{50}^{1/6} = 0.009\ s.m^{-1/3}$. Hence, the Manning value employed in the computations is above the minimum value for physically realistic hydraulic resistance. In other words, grain roughness is found to form about 70% of the total roughness and bedform roughness about 30%.

**References**

CAMAE: Problemas que afectan la Navegabilidad en el Río Guayas, Informar, pp. 4–7, 2013.

CELEC: Revista 25 Años de la presa Daule - Peripa, Tech. rep., CELEC EP-HIDRONACIÓN, 2013.

Henderson, F. M.: Open Channel Flow, Macmillan series in civil engineering, Macmillan, New York, 1966.

INEC: Guayaquil en cifras, http://www.ecuadorencifras.gob.ec/guayaquil-en-cifras/, 2017.

Strickler, A.: Beiträge zur Frage der Geschwindigkeitsformel und der Rauhigkeitszahlen für Ströme, Kanäle und geschlossene Leitungen, Communications de l'Office fédéral de l'économie hydraulique, Mitteilungen des eidgenossischen Amtes für Wasserwirtschaft, Bern, No.16, 1923.

**Response to the comments after the revision made by Reviewer 1: Mick van der Wegen**

We would like to thank the Reviewer for the time spent revising the manuscript and the detailed comments provided. Please find below responses to the each of your remarks about the manuscript in a comment-by-comment basis.

**R1-Comment 1: My first comment was related to the clarity of the methodology and you did not provide a response (" 1) Your methodology is as follows . 1-derive a 'realistic' bathymetry starting from a flat bed. 2- validate hydrodynamics-sediment dynamics and morphodynamics. 3- evaluate model results and scenarios. Update your methodology section so that it reads like this (especially wrt the morphodynamic validation)"). Maybe you overlooked. I still find that you do not explain your methodology clearly in the introduction. I think that it should be made clear just after the aim of the paper otherwise it only becomes clear at the end of the paper what you did (conclusions) which should not be the case.**

According to the reviewer suggestion we added the following text in p.4 line 28:

2.1 Outline

A process-based numerical model was implemented in order to reproduce the morphological development of the Guayas River and the processes behind its evolution. The implementation of the model requires data of different nature that describe the boundary conditions and the geometry of the river. Among these data, the bed topography poses a particular problem, since information in a suitable resolution is lacking. Therefore, prior the analysis, the derivation of a realistic initial bed topography was performed with the aid of a long-term morphodynamic simulation. The respective validation of hydrodynamics, sediment dynamics and morphodynamics followed. Finally, according to the major developments carried out in the estuary, a number of scenarios was defined. The analysis is focused on evaluating the effects on the sediment budget of each of the individual scenarios, by comparison with a reference case that aims to mimic the actual situation.

**R1-Comment 2: My second comment was related to a description of the initial bathymetry generation; You responded by a clear description of how the initial bathymetry was generated, but I fail to see this description in the manuscript. I think it is essential and that you have to include it since it is a crucial aspect of your work. Also this publication should be self explanatory..... "Available journal space" should not be the leading argument here. The text could read (based on your explanation text):**

According to the reviewer suggestion we added the following text in p.10 line 11:

In relation to morphodynamics, since bed level data were lacking for a significant part of the inner estuary, a long-term morphological simulation was performed in order to derive the missing information, as mentioned in subsection 2.1. In that regard, the topography for the entire Guayas River was initially set as a flat bed. A corresponding initial level of 6.00 m below mean sea level was determined based on the theory posed by Savenije (2006), that describes a general equilibrium state for alluvial estuaries in which the mean depth and the tidal amplitude remain constant along the estuary. The model then is run until some stable

patterns are generated. The development of the estuary's depth (averaged over the domain) was used to assess the stability condition. In total, the morphodynamic simulation took about 200 years to reach the equilibrium, i.e., when the evolution of the estuary's depth remains constant. The obtained topography was contrasted against the few areas where information was available. It could then be verified that some characteristic observable features such as the formation of the "El Palmar Islet" were properly captured by the generated bed. As a final stage to validate the topography, the computed water levels were compared with those pertaining to measurements at the Guayaquil tidal station over a spring-neap tidal cycle. A similar model performance in relation to the results of the other tidal stations could be achieved. In addition, after performing a tidal analysis for both measured and modeled water levels, a generally good agreement could be verified for the amplitude and phases of the most energetic components. See Barrera Crespo (2016) for more detailed information.

**References**

Barrera Crespo, P. D. (2016). "Delft3D Flexible Mesh modelling of the Guayas River and Estuary system in Ecuador." Delft University of Technology, National University of Singapore, Delft. (http://resolver.tudelft.nl/uuid:c8a4c2f1-208b-4332-a17f-8afb28ec71e6).

Savenije, H. H. G. (2006). Salinity and tides in alluvial estuaries. Elsevier.

**Response to the comments made by Reviewer 2: Dušan Žagar**

We thank the Reviewer for the revision of the manuscript. Please find below responses to the each of your remarks in a comment-by-comment basis.

**R2-Comment 1: P9-line15: please replace "somewhat constant" with a more suitable formulation; something (tidal amplitude) can be either constant or variable..**

According to the reviewer suggestion we replaced the text "somewhat constant" with "constant" in p.10 line 10.